# Exploring the Role of the Environment as a Reservoir of Antimicrobial-Resistant *Campylobacter*: Insights from Wild Birds and Surface Waters

**DOI:** 10.3390/microorganisms12081621

**Published:** 2024-08-08

**Authors:** Louise Hock, Cécile Walczak, Juliette Mosser, Catherine Ragimbeau, Henry-Michel Cauchie

**Affiliations:** 1Environmental Research and Innovation (ERIN) Department, Luxembourg Institute of Science and Technology (LIST), 41 Rue du Brill, L-4422 Belvaux, Luxembourg; cecile.walczak@list.lu (C.W.); henry-michel.cauchie@list.lu (H.-M.C.); 2Epidemiology and Microbial Genomics, Laboratoire National de Santé (LNS), 1 Rue Louis Rech, L-3555 Dudelange, Luxembourg; catherine.ragimbeau@lns.etat.lu

**Keywords:** *Campylobacter*, antimicrobial resistance, resistome, virulome, MLST, wild bird, surface water

## Abstract

Antimicrobial resistance (AMR) is a growing global health challenge, compromising bacterial infection treatments and necessitating robust surveillance and mitigation strategies. The overuse of antimicrobials in humans and farm animals has made them hotspots for AMR. However, the spread of AMR genes in wildlife and the environment represents an additional challenge, turning these areas into new AMR hotspots. Among the AMR bacteria considered to be of high concern for public health, *Campylobacter* has been the leading cause of foodborne infections in the European Union since 2005. This study examines the prevalence of AMR genes and virulence factors in *Campylobacter* isolates from wild birds and surface waters in Luxembourg. The findings reveal a significant prevalence of resistant *Campylobacter* strains, with 12% of *C. jejuni* from wild birds and 37% of *C. coli* from surface waters carrying resistance genes, mainly against key antibiotics like quinolones and tetracycline. This study underscores the crucial role of the environment in the spread of AMR bacteria and genes, highlighting the urgent need for enhanced surveillance and control measures to curb AMR in wildlife and environmental reservoirs and reduce transmission risks to humans. This research supports One Health approaches to tackling antimicrobial resistance and protecting human, animal, and environmental health.

## 1. Introduction

With 4.95 million deaths associated with drug-resistant bacterial infections in 2019, antimicrobial resistance (AMR) is considered as the new silent pandemic [1]. The emergence of AMR represents a complex interplay of factors, including the overuse and misuse of antimicrobials in human and veterinary medicine, as well as in agricultural practices [2]. This selective pressure drives the evolution of antimicrobial-resistant bacteria (ARB) by promoting their survival and proliferation [3]. With the emergence of resistant strains, different risks of dissemination can occur, ranging from the spread of the acquired genes through horizontal gene transfer or mobile genetic elements to the geographic dispersion of the resistant clone itself thanks to international travel and trade [4]. AMR poses a significant threat to global public health, challenging the efficacy of antimicrobials and complicating the treatment of infectious diseases [5], contributes to increased morbidity, mortality, and healthcare costs [6], and jeopardises the success of medical procedures such as organ transplantation, chemotherapy, and surgery [7]. In addition to its direct impact on human health, AMR also compromises animal health and welfare, affecting food production systems and posing challenges to sustainable agriculture [8].

Environmental reservoirs, including surface waters, soil, and wildlife habitats, serve as vast repositories for diverse microbial communities, providing ample opportunities for the exchange and acquisition of resistance genes [9]. Factors such as agricultural runoff, the improper disposal of antimicrobial residues, and the discharge of untreated sewage contribute to the contamination of environmental matrices with antimicrobial agents and ARB [10]. Once introduced into these ecosystems, ARB can persist, survive, and sometimes proliferate, with a potential risk of the transmission of resistant microorganisms to human and animal populations [11]. As such, AMR is a global One Health challenge, involving the transfer of bacteria and genes between humans, animals, and the environment [12].

Zoonotic ARB present in animals and food can, consequently, also compromise the effective treatment of infectious diseases in humans [13]. Among zoonotic bacteria, *Campylobacter* spp. represents a significant bacterial pathogen with substantial implications for both animal and human health. In Europe, *Campylobacter* has been the leading cause of bacterial gastroenteritis since 2005, representing more than 60% of all reported cases in 2022 [14]. The highest notification rate in 2022 was observed in Luxembourg (141.3 cases per 100,000) [14]. However, the true incidence of campylobacteriosis may be higher due to underreporting, diagnostic challenges, and asymptomatic infections [15]. Therefore, *Campylobacter* infections pose a considerable burden on public health systems, causing symptoms ranging from mild diarrhoea to severe abdominal pain, and occasionally leading to complications such as Guillain–Barré syndrome [16]. The virulence of *Campylobacter* strains depends on various factors, including bacteria motility, adhesion to the intestinal mucosa, the invasion of epithelial cells, toxin production, and protein secretion [17]. Campylobacteriosis is primarily associated with the consumption and cross-contamination of contaminated food, particularly poultry products. However, these bacteria are also prevalent in the environment and can colonise the gastrointestinal tracts of various animals, including poultry, cattle, and wild birds [18]. 

Despite the growing recognition of the importance of environmental reservoirs in the transmission of AMR, there remains a paucity of data regarding the prevalence and distribution of resistant *Campylobacter* spp. in wild birds and surface waters. This study aims to address this knowledge gap by investigating the virulence and AMR profiles of *Campylobacter* isolates obtained from wild birds and surface waters in Luxembourg. By elucidating the dynamics of AMR dissemination in these ecological niches, this research will provide valuable insights into the potential role of the environment as a reservoir of ARB and antimicrobial resistance genes (ARGs).

## 2. Materials and Methods

### 2.1. Campylobacter Isolates Collection

A total of 263 *Campylobacter* isolates recovered by the passive filtration method from wild bird faeces and surface waters collected between 2019 and 2021 in Luxembourg [19] were included in the study: 119 *C. jejuni* (110 isolated from wild birds and 9 isolated from surface waters) and 144 *C. coli* (2 isolated from wild birds and 142 isolated from surface waters) (Appendix A). All strains were inoculated on chocolate agar plates with Vitox (PO5090A, Oxoid, Basingstoke, UK) or mCCDA plates (PO5091A, Oxoid, Basingstoke, UK) and incubated over 48 h at 42 °C under microaerobic conditions using the CampyGen gas-generating system (DN0025, Oxoid, Basingstoke, UK).

### 2.2. Disk Diffusion Method

AMR was assessed using the disk diffusion method on Mueller–Hinton agar with 5% sheep blood (PB0431, Oxoid, Basingstoke, UK) for various classes of antibiotics. For aminoglycosides, gentamicin (CN, 10 µg) (CT0024B, Oxoid, Basingstoke, UK) was used. Beta-lactam resistances were tested with ampicillin (AMP, 10 µg) (CT0003B, Oxoid, Basingstoke, UK) and amoxicillin/clavulanic acid (AMC, 20/10 µg) (CT0223B, Oxoid, Basingstoke, UK). Quinolone resistances were evaluated using nalidixic acid (NAL, 30 µg) (CT0031B, Oxoid, Basingstoke, UK) and ciprofloxacin (CIP, 5 µg) (CT0425B, Oxoid, Basingstoke, UK). Macrolide resistances were assessed with erythromycin (ERY, 15 µg) (CT0020B, Oxoid, Basingstoke, UK). Resistance to phenicols was tested with florfenicol (FFC, 30 µg) (CT1754B, Oxoid, Basingstoke, UK). Finally, tetracycline resistance (TET, 30 µg) (CT0054B, Oxoid, Basingstoke, UK) was also considered. The testing followed the French Microbiology Society (SFM) and EUCAST recommendations (Recommendations 2020 v1.1 April) [20]. The breakpoints were set as follows: AMP, 14 mm; AMC, 14 mm; CIP, 26 mm; ERY, 20 mm; and TET, 30 mm [20]. Additionally, according to Tang et al. [21], the breakpoints for NAL and FFC were 32 mm and 16 mm, respectively.

### 2.3. Genomic DNA Extraction and Whole-Genome Sequencing

DNA was extracted by using the QIAamp DNA Mini Kit (Qiagen, Venlo, The Netherlands) according to the manufacturer’s instructions. DNA was quantified with the Qubit^®^ 2.0 Fluorometer (Invitrogen, Merelbeke, Belgium) and the Qubit^®^ dsDNA HS Assay kit (Life Technologies, Gistel, Belgium). The DNA concentration was adjusted to be within the range between 3 and 17 ng/µL for subsequent sequencing. Libraries were prepared using the Nextera™ DNA Flex Library Prep Kit (Illumina, San Diego, CA, USA) and sequenced on the MiSeq™ platform (Illumina, USA), achieving 250-bp paired-end reads. The datasets of the sequence raw reads used for this study can be found in the ENA projects PRJEB57730 and PRJEB75211.

### 2.4. Genomic Assembly and Characterisation

The paired-end raw read data were de novo assembled using Spades v.3.11.1 (default parameters) implemented on Ridom SeqSphere+ v8.3.1 (Ridom GmbH, Münster, Germany) [22]. The Multi-locus sequence typing (MLST, 7 loci) [23] and core genome MLST (cgMLST, 637 loci) was assigned by using a scheme available in Ridom SeqSphere+, and isolates were classified in Sequence Type (ST). NCBI AMRFinderPlus v3.10.5 [24] (database v2021-06-01.1), deployed with the Ridom SeqSphere+ installation, was used to find genes and point mutations related to AMR, biocide, stress resistance, and virulence (BLAST identity > 90% and aligned = 100%).

### 2.5. Result Interpretation and Statistical Analyses

Multidrug-resistant (MDR) bacteria are defined as those displaying resistance to at least three classes of antibiotics. Measurements of proportion and their 95% confidence intervals (CI95s) were determined by following a binomial law approximated by a normal law (Wilson method). The concordance rate between the AMR genotype and the phenotype refers to the correspondence between the observed phenotypic antibiotic resistance and the presence of resistance genes or mutations associated with that phenotype. The concordance rate was calculated as the number of isolates with a matching phenotype and genotype (the difference between the total number of isolates and the difference between the number of resistant genotypes and resistant phenotypes) divided by the total number of isolates tested. Cohen’s kappa coefficient was used to determine the agreement between the pairs of phenotypes and genotypes of AMR and resistance determinants for all isolates. The interpretation of the kappa coefficient, expressed as the strength of agreement, was: <0.00: poor; 0.00–0.20: slight; 0.21–0.40: fair; 0.41–0.60: moderate; 0.61–0.80: substantial; and 0.81–1.00: almost perfect [25]. The association between AMR and other characteristics (bird species, upstream/downstream from a wastewater treatment plan, arsenic resistance) was determined using the chi-squared test (Rstudio v2022.02.0+443), with a probability value of *p* < 0.05 considered to be statistically significant. UPGMA trees were constructed by pairwise analyses of alleles of MLST and cgMLST, with missing targets ignored using the default settings (Ridom SeqSphere+ v8.3.1). UPGMA trees for virulome analyses were constructed by a pairwise analysis of 126 virulence genes, with missing values considered as their own category (Ridom SeqSphere+ v8.3.1).

## 3. Results

### 3.1. AMR Phenotypes and Genotypes of Isolates

The *Campylobacter coli* and *C. jejuni* isolates were analysed for their resistance to eight antibiotics using the disk diffusion method. *C. coli* were mainly isolated from surface waters and 3% were resistant, whereas *C. jejuni* were mainly isolated from wild birds and 12% were resistant to at least one antibiotic (CN, AMP, AMC, NAL, CIP, ERY, FFC, and TET). One *C. jejuni* isolate, from western jackdaw, was resistant to nalidixic acid, ciprofloxacin, and tetracycline. In surface waters and wild birds, *Campylobacter* showed resistance to beta-lactams, quinolone, and tetracycline (Figure 1, Table 1).

Genotypically, genes or mutations conferring AMR were detected for aminoglycosides, beta-lactams, quinolones, macrolides, and tetracycline (Figure 2, Table 2). In surface waters, 48% of *C. coli* isolates possessed resistance genes corresponding to at least one antibiotic, and one isolate was MDR (aminoglycoside, beta-lactam, and quinolone). Most resistant genotypes possessed a mutation responsible for aminoglycoside resistance (36.6% of isolates) or genes conferring beta-lactam resistance (13.4% of isolates). Among the nine *C. jejuni* isolates from surface waters, six showed resistance genes to beta-lactam and two possessed resistance genes to beta-lactam and tetracycline and mutations conferring resistance to quinolone. In wild birds, 89% of isolates possessed resistance genes to beta-lactam, where, among them, 11% also possessed mutations conferring resistance to quinolone, aminoglycoside, or tetracycline and one isolate possessed an MDR genotype (beta-lactam, macrolide, and tetracycline). Another MDR profile was isolated from birds with genes conferring tetracycline resistance and mutations conferring macrolide and quinolone resistance.

The concordance of resistant genotypes (predicted by AMRFinder from whole-genome sequencing data) and phenotypes (confirmed by disk diffusion) was between 98 and 100%, consistent with predictions observed for the database validation [24], except for the genes aadE-Cc detected by AMRFinder, but without a resistant phenotype observed for aminoglycoside and some *bla_OXA_* genes which could confer beta-lactam resistance (Table 3). This concordance was verified by the Cohen’s kappa coefficient measuring the inter-rater reliability, which confirmed a slight concordance between phenotypes and genotypes for beta-lactam resistance in *C. coli* and *C. jejuni* (Table 4).

Macrolide resistance was associated with an A103V mutation in the ribosomal protein L22 gene. ARG *tet(O)* was associated with tetracycline resistance. *aadE* genes coding for aminoglycoside-modifying enzymes were detected and conferred aminoglycoside resistance. Point mutations in the ribosomal protein P12 (RpsL) acting as a streptomycin-interacting residue were also detected. A *gyrA* mutation resulting in the amino acid substitution T86I was associated with resistance to (fluoro)quinolones (ciprofloxacin and nalidixic acid). Nineteen variants of the *bla_OXA_* genes involved in beta-lactam resistance were detected in 126 isolates (Appendix A). Nine isolates were phenotypically ampicillin-resistant, and six of them (67%) also carried at least one gene coding for a beta-lactamase of the OXA-like family.

### 3.2. Association between AMR, ST, and Heavy Metals

An association was established between STs and the AMR profile resulting from both analyses, WGS in silico prediction, and antibiograms (Figure 3). All isolates assigned to the same ST showed a similar resistance profile to the same antibiotic classes, with two exceptions: *C. coli* ST 1766 and ST 1981. Only one of three isolates from ST 1766 expressed resistance to AMP and only two of seven isolates from ST1981 were in silico resistant to aminoglycosides.

Two STs exhibited MDR profiles in silico only: ST 10042 (*C. coli*), associated with streptomycin, quinolone, and beta-lactam (*bla_OXA-193_*) resistances, and ST 10815 (*C. jejuni*), harbouring erythromycin, tetracycline, and beta-lactam (*bla_OXA-637_*) resistances determinants. In addition, two *C. jejuni* isolates from ST 9897 were classified as MDR in silico and phenotypically for the following antibiotic classes: quinolone (NA and CIP, *gyrA_T86*), beta-lactam (*bla_OXA-193_*), and tetracycline (*tet(O)*). These two strains were isolated on the same day, but at two different places (distance of around 10 km) in the Alzette river. *C. jejuni* ST 11004 was the only isolate possessing a mutation for streptomycin resistance, but without phenotypical resistance against gentamycin. 

The bird species of the isolates showed no relation with the AMR pattern. Likewise, no more resistant isolates were detected in surface water after a wastewater treatment plan (WWTP). However, MDR strains were only detected after a WWTP (Appendix A). Additional genes conferring arsenic resistance, *arcP* and *arc3*, were also detected in our isolates. These arsenic resistance genes were significantly more prevalent in *C. jejuni* than *C. coli* (Figure 4). However, the presence of heavy metal tolerance genes was not correlated with the presence of ARGs.

### 3.3. Virulence Genes of Isolates

The genomic analysis of 263 *C. coli* and *C. jejuni* isolates revealed, in total, 119 virulence-associated genes (70 and 114, respectively) related to motility, chemotaxis, adhesion, and invasion (Table 5). All the isolates harboured the genes *flgB*, *flgC*, *flgF*, *flgG*, *flgI*, *flhA*, *fliE*, *fliF*, *fliG*, *fliI*, *fliL*, *fliM*, *fliS*, and *fliW*, all involved in the flagellar structure, *pseB* and *flaC* involved in flagellin synthesis and consequently implicated in the adhesion and invasion of host cells, and *hldD* involved in lipooligosaccharide (LOS) synthesis. 

Several virulence factors (*n* = 49) were only present in *C. jejuni* isolates. The *cadF* gene encodes an outer membrane protein mediating the specific binding of *C. jejuni* to fibronectin, promoting *C. jejuni* adhesion. *Campylobacter* invasion antigens (Cia), required to invade host cells, were also present in all the *C. jejuni* isolates. The *wlaN* gene, associated with Guillain–Barré syndrome [26], was identified in six (5.5%) *C. jejuni* isolates from wild birds. *cdtABC* genes, coding for cytolethal distending toxin (CDT) production, were only present in 47% (56 out of 119) of *C. jejuni* isolates. Only one *C. coli* harboured the *pVir* plasmid-encoding genes (*Cjp54*, *virB11*, *virB4*, *virB8*, and *virD4*) homologous to the type IV secretion system found in *Helicobacter pylori* [27]. The number of virulence-associated genes was higher in *C. jejuni* than in the *C. coli* isolates (Appendix A). The strains with the most detected virulence genes belonged to ST 19 and ST 464. The dendrograms in Appendix A group isolates based on the similarity of their virulomes. Generally, isolates from the same ST corresponded to a cluster according to their virulome, except *C. jejuni* ST 448 and *C. coli* ST 1981, 11402, 12026, and 12033.

## 4. Discussion

Wild birds, and more generally wildlife, are recognised as vectors and spreaders for a wide range of microorganisms, including *Campylobacter*, which is transmissible to other animals and humans [17]. An additional risk factor linked to *Campylobacter* environmental reservoirs is the spread of their AMR. *Campylobacter* spp. stands as a notable bacterial pathogen contributing significantly to the AMR crisis. Previous studies have shown that wild birds, including migratory species, can harbour *Campylobacter* strains with various AMR profiles [28,29,30]. Surface waters, acting as repository for faecal contaminants from diverse sources, have also been implicated in the dissemination of AMR, with *Campylobacter* isolates, exhibiting resistance to multiple antimicrobials, frequently detected in these environments [31]. Hence, elucidating the prevalence and distribution of ARGs in *Campylobacter* isolated from the environment, including wild birds and surface waters, is crucial in understanding the dynamics of AMR dissemination and elaborating effective mitigation strategies.

In our study, the main isolated species in wild birds was *C. jejuni*, on the contrary to surface waters, where *C. coli* was predominant, in accordance with previously reported studies [29,32]. Mutations in the ribosomal protein L22 and *gyrA*, as well as the presence of *tet(O)* and *bla_OXA_* genes, were identified as the prevalent mechanisms conferring macrolide, quinolone, tetracycline, and beta-lactam resistance, respectively, and in accordance with other global observations [26,33,34,35]. In surface waters, almost half of the *C. coli* isolates presented resistance genes to at least one antibiotic. Resistances to aminoglycosides (36.6%) and beta-lactams (13.4%) were the most observed. In wild birds, almost 90% of the isolates possessed ARGs to beta-lactams, which is similar to the proportion of beta-lactam-resistant *C. jejuni* in turkeys, according to a recent study [26]. This antibiotic class is not typically prescribed for treating campylobacteriosis and is rarely used for routine monitoring [34]. Most cases of *Campylobacter* gastroenteritis do not require treatment, as they are generally short-lived, self-limited events. However, when symptoms are prolonged or very severe, common antimicrobial agents prescribed are macrolides, such as erythromycin, and fluoroquinolones, such as ciprofloxacin [36]. Macrolides are classified as critically important antibiotics by the World Health Organisation (WHO), and fluoroquinolones as the highest priority critically important antimicrobials because of their consideration as a last resort of sole therapy for treating serious MDR infection in humans [37]. The WHO even considers fluoroquinolone-resistant *Campylobacter* spp. as a high-priority pathogen [38]. However, high to extremely high levels of resistance to ciprofloxacin and tetracycline were reported in human *C. jejuni* and *C. coli* [39]. Erythromycin resistance for *C. jejuni* was not detected or was very low, but was higher in *C. coli*, and combined resistance with ciprofloxacin was rare [39]. Resistance genes against antibiotics commonly used to treat campylobacteriosis, macrolides, and fluoroquinolones were observed in 5% of all our isolates. 

While our study identified a diverse array of known AMR genetic determinants in *Campylobacter* isolates from wild birds and surface waters, the observed phenotype resistance was not necessarily equated. In our study, only 3% of *C. coli* isolates from surface water and 12% of *C. jejuni* isolates from wild birds expressed resistance. The proportion of susceptible isolates was higher than that in other European and non-European studies, e.g., 30% of *C. coli* isolates from surface waters and 67–78% of *Campylobacter* isolates from wild birds were resistant [29,32,40]. All resistant *C. coli* isolates in this study exhibited resistance to ampicillin, constituting 3% of the total *C. coli* isolates. Similarly, 3% of *C. jejuni* isolates from birds were resistant to ampicillin. Among the *C. jejuni* isolates from birds, 8% were resistant to quinolones (NAL and CIP), and 1% demonstrated resistance to both quinolones and tetracycline (NAL, CIP, and TET). Our findings indicated slightly lower resistance rates compared to previous studies, which reported 11.1–12.5% resistance to quinolones and 6.0–22.2% resistance to both quinolones and tetracycline [40,41]. This difference may be attributed to the fact that most *Campylobacter* strains in Luxembourg water are associated with wild birds rather than poultry, probably due to the low poultry and slaughtering intensity in the country [42]. It is expected that wild bird strains are not under antimicrobial selective pressure and so exhibit lower resistance compared to poultry isolates, explaining the lower resistance rate observed in the Luxembourg environment. 

The concordance between genome-based predicted and phenotypic AMR profiles for quinolone, macrolide, and tetracycline underscores the reliability of whole-genome-sequencing-based approaches for defining resistances in *Campylobacter* isolates [24,43]. Nevertheless, this concordance was low for beta-lactams and aminoglycosides. Using an automated annotation pipeline to detect ARGs could mislead some mechanisms. The phenotypic resistance profile could be impacted by transcriptional regulation, the presence or absence of efflux pumps and their expression, membrane permeability, some frameshift mutations, and/or the existence of mosaic or new resistance genes [26,44]. Some technical issues, such as poor-quality sequences, assembly errors, and/or incorrect analyses or incomplete databases, may also contribute to discrepancies in genomic and phenotypic AMR profiles [45,46]. In addition, our phenotypic resistance profiles were observed by the disk diffusion method and analysed according to cut-off values relevant in clinical applications. Other methods like broth microdilution (MIC) or the E-test (BioMerieux, Craponne, France) could highlight the lower resistance seen in genotypes. 

In the present study, most of the *bla_OXA_* genes did not correspond to phenotypic resistance to ampicillin. This observation that the majority of *Campylobacter* isolates harbouring the *bla_OXA-61_* gene were still susceptible to ampicillin was previously reported [47,48]. A single-nucleotide G-T transversion in the *bla_OXA-61_*-like promoter area is responsible for the expression level of beta-lactamase and, thus, ampicillin resistance [34]. In our study, only one strain harbouring the *bla_OXA-61_* gene was confirmed as being resistant to ampicillin, whereas the 15 others remained susceptible. Therefore, further investigations are required to understand the relationship between genetic determinants of resistance and the environmental factors shaping antimicrobial susceptibility to refine our strategies for predicting and mitigating the emergence and spread of AMR, including in environmental reservoirs. 

Excluding *bla_OXA_* genes, 25.5% of the isolated *Campylobacter* possessed ARGs: 12% of *C. jejuni* from wild birds and 37% of *C. coli* from surface waters had genes known to confer AMR. These proportions align with our phenotype observations, with a 97% concordance rate. In Luxembourg, 62% and 82% of *C. jejuni* and *C. coli* human isolates, respectively, are resistant to at least one antibiotic agent (CN, AMC, CIP, ERY, and TET) [49]. Through wastewater treatment plants, resistant *Campylobacter* can be introduced into the environment. Although the proportion of resistant *Campylobacter* isolated from wild birds and surface water was lower than that in human isolates, it remains a concern, as there is a risk of persistence, spread, and sources of contamination over time. 

In our study, 2% of isolated *C. coli* and *C. jejuni* strains (*n* = 5) harboured at least three resistance genomic determinants and can be considered as MDR *Campylobacter*. Notably, in surface waters, MDR strains (*n* = 3) were only detected downstream from a WWTP. Two *C. jejuni* isolates, identified as MDR both in silico and phenotypically (quinolone, beta-lactam, and tetracycline), sharing the same ST (ST 9897), were collected on the same day, from the same river, at locations 10 km apart. Additionally, ST 9897 has largely been found in human stools in the United Kingdom since 2021, according to the PubMLST database [50]. This finding reinforces the hypothesis that *Campylobacter* strains, including resistant ones, have a significant spreading capacity, with water acting as a vehicle for the geographic dispersion of AMR. Surface waters play a crucial role in transmitting *Campylobacter* among humans, farm animals, and wild animals, including wild birds, through close contact [32]. Additionally, bird migration facilitates the establishment of disease far from the original infection site. Furthermore, some bird species that have adapted to anthropogenic environments can easily transmit zoonotic diseases through close contact with livestock, domestic animals, and humans in urban and agricultural settings [51]. A previous study reported that generalist and recurring lineages of *C. jejuni* found in infected patients are also present in several species of wild birds [19], highlighting their potential as reservoirs for human infection. These findings also emphasise the role of the water cycle in collecting clinical strains from patients through wastewater, which then disperse into surface waters, contaminating both wild and food animals. 

The presence of virulence-associated genes in environmental *Campylobacter* genomes has also been studied. Genes encoding adhesion, invasion, and cytotoxin production underscore the adaptability of environmental *Campylobacter* to colonise and cause disease in hosts, including humans. The number of virulence-associated genes was higher in *C. jejuni* than in *C. coli* isolates, concurring with other recent observations [33,52,53]. Nevertheless, this conclusion may be biased, because the virulome of *Campylobacter* spp. has mainly been studied in *C. jejuni*, and most of the commonly used virulence factor databases were constructed based on the reference genome of this species. Consequently, the lower number of virulence factors detected in *C. coli* may be attributed to either the absence of *C. coli*-specific genes in the queried database or a sequence identity between *C. coli* virulence genes and those in the *C. jejuni* reference genomes below the VFDB tool threshold [33]. All the isolates harboured genes implicated in the flagellar structure and LOS (lipooligosaccharide) synthesis. The flagella contribute to the natural motility of *Campylobacter* required to move into the mucus layer covering the intestinal epithelial cells and colonise the intestine [54]. LOS is located on the *Campylobacter* surface and is involved in the adhesion and invasion of epithelial cells. Moreover, the mimicry between the LOS structure of some *C. jejuni* strains and the neuronal gangliosides can cause a cross-reactive antibody response leading to Guillain–Barré syndrome [54]. A previous study described that specific virulence genes were associated with particular STs [33]. In our study, the genomic diversity was high and did not allow for associating an ST with a specific virulome.

The AMR of *Campylobacter* may not be directly linked to the degree of virulence the strains exhibit, consistent with a previous study [33]. Despite possessing fewer virulence genes, *C. coli* exhibited a higher proportion of antimicrobial resistance genes. Additionally, no correlation was observed between bird species and AMR profiles, nor between heavy metal tolerance genes and ARGs, contrary to the hypothesis of synergistic co-resistance between heavy metals and antimicrobials [55]. 

In conclusion, the use of WGS is invaluable for the monitoring and epidemiology of *Campylobacter* AMR. WGS offers accurate predictions of AMR phenotypes, while also providing detailed insights into their genetic determinants and broader genomic context [56]. Simultaneously, WGS enables the traceability of infections by genomic comparisons of different strains, thereby identifying potential transmission pathways. Our study further highlights that a rapid analysis to determine the ST of *Campylobacter* could reliably infer the AMR characteristics of strains if the database includes previously analysed STs. This finding underscores the potential of WGS and rapid ST determination as powerful tools for tracking and managing AMR in microbial populations.

The proportion of resistant *Campylobacter* isolates observed in our study is noticeable, since wild birds, unlike domestic animals and humans, are not directly exposed to antimicrobials. In particular, the main resistances observed in these environmental niches were against antibiotics commonly used in human and veterinary medicine: macrolides and quinolones. Our findings point out the key roles of wild birds and surface waters as environmental reservoirs and vehicles for spreading resistant *Campylobacter*. Moreover, migratory bird carriers can disperse ARGs or MDR bacteria over large distances. Consequently, the environment plays a critical role in public health by conveying AMR to wildlife throughout the water cycle. These findings underscore the need for enhanced surveillance and control measures to mitigate the spread of AMR in the environment and reduce the risk of transmission to humans and animals. Embracing a holistic One Health approach that includes the environmental dimension of bacterial infection and AMR is essential to preserve the health of humans, animals, wildlife, and the environment.

## Figures and Tables

**Figure 1 microorganisms-12-01621-f001:**
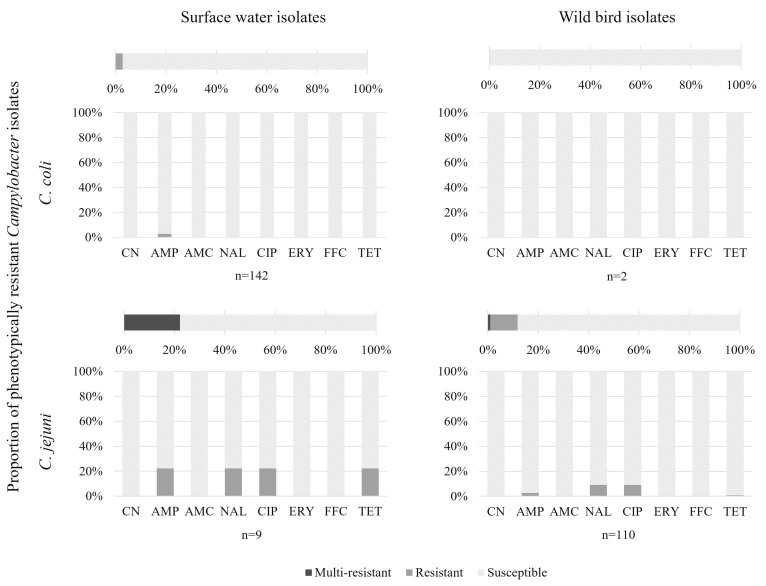
Proportion of phenotypically antibiotic-resistant *Campylobacter* isolates from surface waters and wild birds. CN, gentamycin; AMP, ampicillin; AMC, amoxicillin/clavulanic acid; NAL, nalidixic acid; CIP, ciprofloxacin; ERY, erythromycin; FFC, florfenicol; and TET, tetracycline.

**Figure 2 microorganisms-12-01621-f002:**
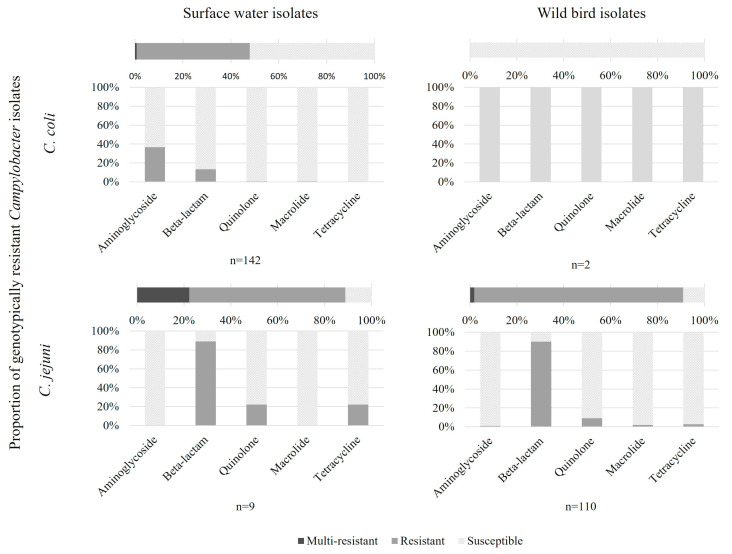
Proportion of genotypically antibiotic-resistant *Campylobacter* isolates from surface waters and wild birds.

**Figure 3 microorganisms-12-01621-f003:**
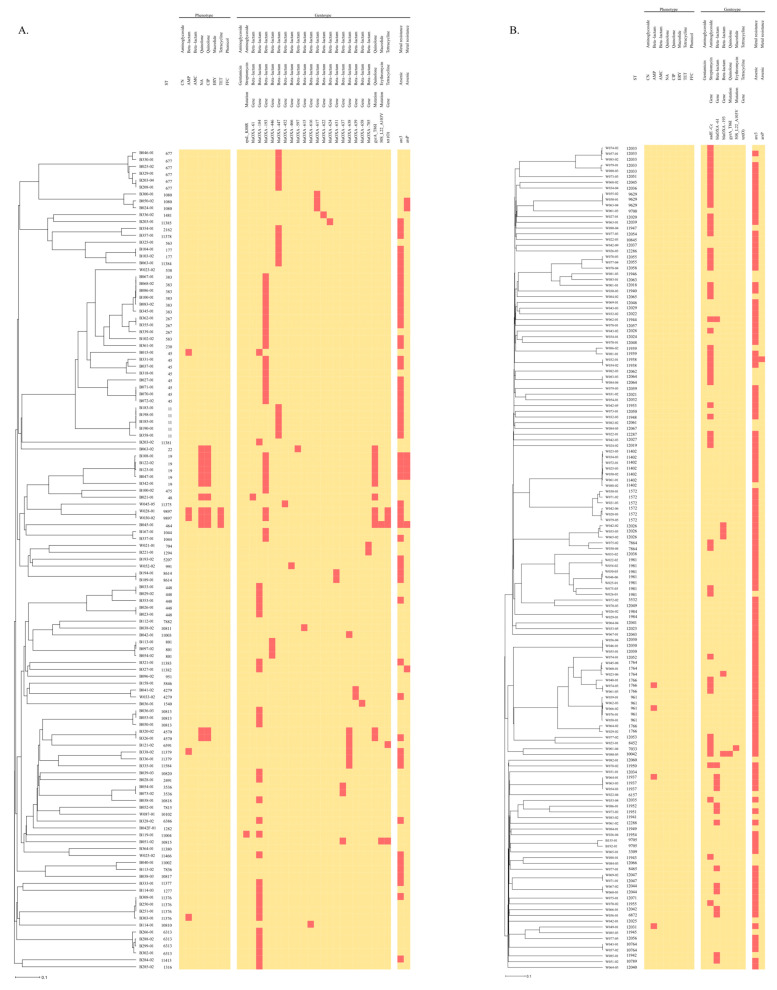
UPGMA dendrogram of cgMLST typing of *C. jejuni* (**A**) and *C. coli* (**B**) strains compiled with heatmap of AMR phenotype and genotype patterns. The red squares indicate the phenotypic resistance or the presence of AMR determinants. ST, Sequence Type.

**Figure 4 microorganisms-12-01621-f004:**
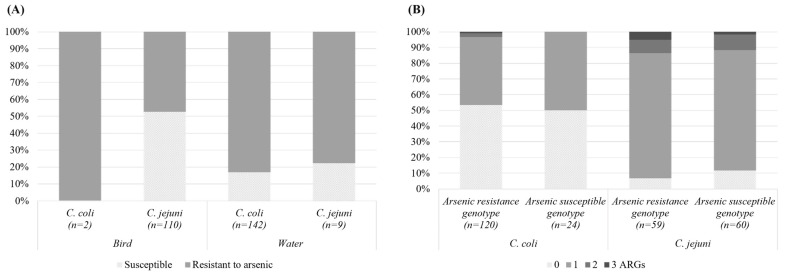
(**A**) Proportion of isolates from wild birds and surface waters possessing resistance gene(s) to arsenic and (**B**) proportion of susceptible *Campylobacter* isolates without ARGs and resistant isolates with one, two, or three ARGs in function of the resistance to arsenic.

**Table 1 microorganisms-12-01621-t001:** Phenotypic antibiotic resistance profiles of *Campylobacter* isolates. AMP, ampicillin; NAL, nalidixic acid; CIP, ciprofloxacin; TET, tetracycline; and CI95, 95% confidence interval.

	*C. coli*		*C. jejuni*	
AMR Profiles	Bird (*n* = 2)	Water (*n* = 142)	Bird (*n* = 110)	Water (*n* = 9)
Susceptible	2 (100%)	138 (97.2%, CI95: 93.0–98.9%)	97 (88.2%, CI95: 80.8–93.0%)	7 (78%)
AMP		4 (2.8%, CI95: 1.1–7.0%)	3 (2.7%, CI95: 0.9–7.7%)	
NAL-CIP			9 (8.2%, CI95: 4.4–14.8%)	
NAL-CIP-TET			1 (0.9%, CI95: 0.2–5.0%)	
AMP-NAL-CIP-TET				2 (22%)

**Table 2 microorganisms-12-01621-t002:** Genotypic antibiotic resistance profiles of *Campylobacter* isolates.

	*C. coli*		*C. jejuni*	
AMR Profiles	Bird (*n* = 2)	Water (*n* = 142)	Bird (*n* = 110)	Water (*n* = 9)
Susceptible	2 (100%)	74 (52.1%, CI95: 43.9–60.2%)	10 (9.1%, CI95: 5.0–15.9%)	1 (11.1%)
Aminoglycoside		48 (33.8%, CI95: 26.5–41.9%)		
Beta-lactam		16 (11.3%, CI95: 7.1–17.5%)	87 (79.1%, CI95: 70.6–85.6%)	6 (66.7%)
Aminoglycoside–Macrolide		1 (0.9%, CI95:0.2–3.9%)		
Aminoglycoside–Beta-lactam		2 (1.2%, CI95: 0.4–5%)	1 (0.9%, CI95: 0.2–5%)	
Beta-lactam–Quinolone			9 (8.2%, CI95: 4.4–14.8%)	
Beta-lactam–Tetracycline			1 (0.9%, CI95: 0.2–5%)	
Aminoglycoside–Beta-lactam–Quinolone		1 (0.9%, CI95: 0.2–3.9%)		
Beta-lactam–Macrolide–Tetracycline			1 (0.9%, CI95: 0.2–5%)	
Beta-lactam–Quinolone–Tetracycline				2 (22%)
Macrolide–Quinolone–Tetracycline			1 (0.9%, CI95: 0.2–5%)	

CI95, 95% confidence interval.

**Table 3 microorganisms-12-01621-t003:** Concordance between resistant genotypes (AMRFinder) and phenotypes (disk diffusion) of isolates.

		*C. coli*		*C. jejuni*	
		Bird (*n* = 2)	Water (*n* = 142)	Bird (*n* = 110)	Water (*n* = 9)
Aminoglycoside	No. of isolates with R phenotype	0	0	0	0
	No. of isolates with R genotype	0	52	1	0
	Concordance (%)	100	63	99	100
Beta-lactam	No. of isolates with R phenotype	0	4	3	2
	No. of isolates with R genotype	0	19	99	8
	Concordance (%)	100	89	13	33
Quinolone	No. of isolates with R phenotype	0	0	10	2
	No. of isolates with R genotype	0	1	10	2
	Concordance (%)	100	99	100	100
Macrolide	No. of isolates with R phenotype	0	0	0	0
	No. of isolates with R genotype	0	1	2	0
	Concordance (%)	100	99	98	100
Tetracycline	No. of isolates with R phenotype	0	0	1	2
	No. of isolates with R genotype	0	0	3	2
	Concordance (%)	100	100	98	100

**Table 4 microorganisms-12-01621-t004:** Concordance between phenotype and genotype AMR predictions.

		Phenotype: Susceptible		Phenotype: Resistant				
	Antimicrobial	Genotype: Susceptible	Genotype: Resistant	Genotype: Resistant	Genotype: Susceptible	Cohen’s Kappa Coefficient	95% CI	Interpretation
*C. coli*	Aminoglycoside	92	52	0	0	-	-	
	Beta-lactam	122	18	1	3	0.04	−0.11–0.2	Slight
	Quinolone	143	1	0	0	-	-	
	Macrolide	143	1	0	0	-	-	
	Tetracylcine	144	0	0	0	-	-	
*C. jejuni*	Aminoglycoside	118	1	0	0	-	-	
	Beta-lactam	12	102	5	0	0.01	0–0.02	Slight
	Quinolone	107	0	12	0	1	1	Almost perfect
	Macrolide	117	2	0	0	-	-	
	Tetracylcine	114	2	3	0	0.74	0.40–1	Substantial

**Table 5 microorganisms-12-01621-t005:** No. of isolates with virulence genes.

		*C. coli*		*C. jejuni*	
	Virulence Gene	Bird (*n* = 2)	Water (*n* = 142)	Bird (*n* = 110)	Water (*n* = 9)
Flagellin	*flaC*	2	142	110	9
Flagellar proteins	*flgB*, *flgC*, *flgF*, *flgG*, *flgI*	2	142	110	9
Flagellar biosynthesis protein	*flhA*	2	142	110	9
Flagellar protein	*fliE*, *fliF*, *fliG*, *fliI*, *fliL*, *fliM*, *fliS*, *fliW*	2	142	110	9
Lipooligosaccharide (LOS) synthesis	*hldD*	2	142	110	9
Pseudaminic acid synthesis (Flagellin)	*pseB*	2	142	110	9
Chemotaxis protein	*cheW*	2	141	110	9
Flagellar protein	*flgH*, *flgJ*, *flgQ*	2	141	110	9
Flagellar motor protein	*motA*	2	141	110	9
Chemotaxis protein	*cheV*	2	142	109	9
Flagellar biosynthesis protein	*fliR*	2	142	109	9
Chemotaxis protein	*cheY*	2	140	110	9
Flagellar protein	*flhG*	0	142	110	9
Flagellar protein	*flgK*	2	141	109	9
Lipooligosaccharide (LOS) synthesis	*gmhA*	0	139	109	8
Pseudaminic acid synthesis (flagellin)	*pseC*	0	109	110	9
LOS synthesis	*hldE*	2	96	109	9
RNA polymerase factor sigma-54 (Pse)	*rpoN*	2	87	110	9
Pseudaminic acid synthesis (flagellin)	*pseI*	2	65	110	9
Flagellar protein	*flgE*	2	43	110	9
Flagellar protein	*flaD*	2	94	50	6
Pseudaminic acid synthesis (flagellin)	*pseA*	2	40	92	7
Capsular polysaccharide	*kpsT*	2	15	110	9
Chemotaxis protein	*cheA*	0	9	109	8
Flagellar motor protein	*fliN*	0	6	110	9
Flagellar protein	*flgM*	0	5	110	9
Pseudaminic acid synthesis (flagellin)	*pseF*	0	3	110	9
Pseudaminic acid synthesis (flagellin)	*pseG*	0	2	110	9
Flagellar protein	*flaG*	0	1	110	9
Flagellar protein	*flgQ*	0	1	110	9
Outer membrane fibronectin-binding protein	*cadF*	0	0	110	9
Invasion antigen	*ciaB*, *ciaC*	0	0	110	9
Adherence	*eptC*	0	0	110	9
Flagellar protein	*flgA*, *flgP*, *flgR*, *flgS*	0	0	110	9
Flagellar biosynthesis protein	*flhB*, *flhF*	0	0	110	9
Flagellar biosynthesis protein	*fliA*, *fliH*, *fliP*, *fliY*	0	0	110	9
Lipooligosaccharide (LOS) synthesis	*gmhB*	0	0	110	9
Adhesin	*jlpA*	0	0	110	9
Flagella motor protein	*motB*	0	0	110	9
Adhesin	*pebA*	0	0	110	9
Flagellar protein	*pflA*	0	0	110	9
Lipooligosaccharide (LOS) synthesis	*waaC*	0	0	110	9
Capsular polysaccharide	*kpsS*	0	0	109	9
Capsular polysaccharide	*kpsD*, *kpsE*	0	1	109	7
Capsular polysaccharide	*kpsM*	0	2	108	7
Capsular synthesis	*Cj1419c*	2	20	84	7
Cytolethal distending toxin (CDT)	*cdtC*	0	0	99	9
Capsular synthesis	*Cj1420c*	2	21	77	6
Capsule protein	*kpsC*	0	0	87	6
Capsule biosynthesis and transport	*Cj1417c*	0	0	85	7
Lipooligosaccharide (LOS) synthesis	*gmhA2*	0	4	80	4
Capsule synthesis	*hddA*	0	5	78	4
Flagella protein	*pseH*	0	0	72	7
Major outer membrane protein	*porA*	0	1	57	1
Cytolethal distending toxin (CDT)	*cdtA*	0	0	54	3
Cytolethal distending toxin (CDT)	*cdtB*	0	0	53	3
Lipooligosaccharide (LOS) synthesis	*waaF*	0	1	50	4
Lipooligosaccharide (LOS) synthesis	*htrB*	0	0	45	7
Capsule protein	*cysC*	0	0	47	4
Flagella protein	*ptmA*, *ptmB*	0	0	42	8
Flagella protein	*flgD*	0	1	42	6
Lipooligosaccharide (LOS) synthesis	*neuC1*	0	35	10	0
Capsule synthesis	*Cj1416c*	0	0	40	4
Flagella protein	*fliD*	0	1	39	4
Capsule synthesis	*hddC*	0	1	37	1
Motility accessory factor PseE	*pseE maf5*	0	0	32	5
Lipooligosaccharide (LOS) synthesis	*Cj1135*	0	0	23	7
Capsule protein	*Cj1427c*	0	1	24	1
Lipooligosaccharide (LOS) synthesis	*waaV*	0	0	24	1
Flagellin	*flaA*, *flaB*	0	0	15	4
Motility accessory factor PseD	*pseD maf2*	0	0	14	1
Motility accessory factor	*maf4*	0	0	9	2
Capsule protein	*rfbC*	0	1	9	0
Lipooligosaccharide (LOS) synthesis	*neuA1*, *neuB1*	0	0	9	0
Flagellar protein	*fliK*	0	0	8	1
Lipooligosaccharide (LOS) synthesis	*Cj1137c*, *Cj1138*	0	0	7	0
Lipooligosaccharide (LOS) synthesis	*wlaN*	0	0	6	0
Lipooligosaccharide (LOS) synthesis	*Cj1136*	0	0	5	0
Lipooligosaccharide (LOS) synthesis	*cstIII*	0	0	5	0
Capsule synthesis	*fcl*	0	1	4	0
Capsule protein	*Cj1432c*, *Cj1435c*, *Cj1436c*, *Cj1440c*	0	1	1	0
Capsule protein	*glf*	0	1	1	0
Capsule protein	*kfiD*	0	1	1	0
Capsule protein	*Cj1426c*	0	0	1	0
Type IV secretion system protein	*Cjp54*, *virB11*, *virB4*, *virB8*, *virD4*	0	1	0	0

## Data Availability

The datasets of sequence raw reads used for this study can be found in the ENA projects PRJEB57730 and PRJEB75211.

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
