# Peer review of "Exploring the Role of the Environment as a Reservoir of Antimicrobial-Resistant Campylobacter: Insights from Wild Birds and Surface Waters"

_microorganisms, 2024, doi:10.3390/microorganisms12081621_

Round 1
Reviewer 1 Report
Comments and Suggestions for Authors
The manuscript submitted by Hock et al. presents very interesting results on the characterization of Campylobacter strains isolated from water samples of wild birds.
Both the structure and the content of the manuscript are very well cared for. The methodology is well detailed and the results respond to the objectives set in a simple and clear way.
I only have a few minor comments to propose to the authors and they will not take much time:
- Throughout the text I have found the words "antimicrobial" and "antibiotic" used as synonyms, when they are not. Please, to be more correct, use only the word "antimicrobial" that encompasses antibiotics.
- I recommend that the authors use standardized acronyms, such as ARB for "antimicrobial-resistant bacteria" (from line 55) and ARG for "antimicrobial resistance genes" (from line 47). The acronym should be explained at first mention in the text, as the authors have done for AMR on line 31, and then use the acronym alone. Please replace "antimicrobial resistance" with AMR on lines 128, 137, 147, 161, 219, 253, 288, 329, 333, 349, 382, ​​389, and 404. Similarly, use the acronym MDR for "multidrug resistance" from its first mention on line 172 (or even 124).
- L. 102-103. Are the breakpoints established for AMP, AMC, CIP, ERY and TET as published in the 2020 EUCAST tables? If so, please add the reference at the end of the sentence.
- L. 200. The acronym ST is not previously described in the text. Authors can include the meaning of ST here or in the corresponding section of the methodology.
- Figure 3. At the current size it is impossible to read the figure properly. I recommend enlarging it as much as possible (perhaps it should take up a full page).
- L. 301-302. I do not fully understand this statement. If the Campylobacter strains on the surface of the water were related to the strains in wild birds, would they not be the same species? However, the proportion of C. coli and C. jejuni found in water and in birds is completely opposite.
- I think that figure S2 would be interesting to include in the manuscript and not as a supplement, since it describes in a very visual way the resistance to arsenic in the strains studied. The same goes for tables S1 and S2, which show the concordance for each of the classes to antimicrobials, and table S4 for the identification of virulence factors. I think these are data that are too relevant to be left "hidden" in the supplementary material...
Author Response
Many thanks for your comments, they are very helpful to increase the quality of our paper.
Comment 1: Throughout the text I have found the words "antimicrobial" and "antibiotic" used as synonyms, when they are not. Please, to be more correct, use only the word "antimicrobial" that encompasses antibiotics.
Response 1: We agree. Where antibiotic were not correctly used, we changes by antimicrobial.
Comment 2: I recommend that the authors use standardized acronyms, such as ARB for "antimicrobial-resistant bacteria" (from line 55) and ARG for "antimicrobial resistance genes" (from line 47). The acronym should be explained at first mention in the text, as the authors have done for AMR on line 31, and then use the acronym alone. Please replace "antimicrobial resistance" with AMR on lines 128, 137, 147, 161, 219, 253, 288, 329, 333, 349, 382, ​​389, and 404. Similarly, use the acronym MDR for "multidrug resistance" from its first mention on line 172 (or even 124).
Response 2: We agree. Acronym used have been improved.
Comment 3: L. 102-103. Are the breakpoints established for AMP, AMC, CIP, ERY and TET as published in the 2020 EUCAST tables? If so, please add the reference at the end of the sentence.
Response 3: Eucast book have been added in the reference part.
Comment 4: L. 200. The acronym ST is not previously described in the text. Authors can include the meaning of ST here or in the corresponding section of the methodology.
Response 4: The acronym ST mean Sequence Type and was explained in line 120.
Comment 5: Figure 3. At the current size it is impossible to read the figure properly. I recommend enlarging it as much as possible (perhaps it should take up a full page).
Response 5: we agree, we will provide, to the journal, the figure in a good format to be read.
Comment 6: L. 301-302. I do not fully understand this statement. If the Campylobacter strains on the surface of the water were related to the strains in wild birds, would they not be the same species? However, the proportion of C. coli and C. jejuni found in water and in birds is completely opposite.
Response 6: In a previous study [42], cgMLST was used to determine sources of surface water contamination in Luxembourg and Netherlands. In Luxembourg, main source was wild birds, whereas in Netherlands, the main source was poultry. This fact may be explained by the difference in poultry intensity. They isolated both C. jejuni and C. coli. The antimicrobial selection pressure is higher in poultry than in wild birds. It could explain why the surface water isolated in Luxembourg are more susceptible than in other European countries. In addition, we didn't used the same method for isolation. In our case, we didn't do enrichment and we observed that C. coli are more resistant in water [19]. So, in absence of enrichment step, we isolated more C. coli on plate. Our hypothesis is that C. jejuni are also present in surface water but more in kind of dormant stage. We proved that metagenomic increases the recovery of C. jejuni profiles (DOI: 10.3390/microorganisms11010121). So in our study, we have a bias of C. jejuni recovery due to the isolation method (the enrichment step introduce other bias too). Nevertheless, we can based on previous study [42] to confirmed that surface water in Luxembourg are mainly contaminated by wild birds where the antimicrobials selection pressure in lower.
Comment 7: I think that figure S2 would be interesting to include in the manuscript and not as a supplement, since it describes in a very visual way the resistance to arsenic in the strains studied. The same goes for tables S1 and S2, which show the concordance for each of the classes to antimicrobials, and table S4 for the identification of virulence factors. I think these are data that are too relevant to be left "hidden" in the supplementary material...
Response 7: Figure S2, Tables S1, S2 and S4 have been added in the main text.
Reviewer 2 Report
Comments and Suggestions for Authors
The article titled "Exploring the role of the environment as a reservoir of antimicrobial resistant Campylobacter: Insights from wild birds and surface waters" provides valuable insights. It emphasizes the importance of whole-genome sequencing (WGS) in monitoring and understanding Campylobacter antimicrobial resistance (AMR). WGS is crucial in accurately predicting AMR phenotypes and gaining insights into genetic determinants and broader genomic contexts. By comparing genomes of different strains, WGS also facilitates tracing infections and identifying potential transmission pathways. The authors demonstrate that rapid sequence type (ST) analysis of Campylobacter can reliably infer AMR characteristics if the database includes previously analyzed STs. This underscores the potential of WGS and rapid ST determination in managing AMR in microbial populations. The presence of resistant Campylobacter in wild birds, which are not directly exposed to antibiotics, highlights the role of these birds and surface waters as reservoirs and spreaders of resistance, particularly to antibiotics like macrolides and quinolones. Migratory birds can spread AMR genes or multidrug-resistant bacteria over large distances, emphasizing the environmental role in public health through conveying resistance via the water cycle. These findings underscore the need for enhanced surveillance and control measures to mitigate AMR spread and reduce transmission risks to humans and animals. A holistic One Health approach, integrating environmental aspects of bacterial infection and AMR, is essential for the health of humans, animals, wildlife, and the environment. Congratulations to the authors. I have only minor comments.
Line 93; remove ]“(CT0024B, Oxoid, UK)]”
Line 95; remove ] “(CT0223B, Oxoid, UK)]”
Line 98; remove ] “(CT0020B, Oxoid, UK)]”
Line 103; Add Tang et al. “Additionally, according to Tang et al. [20]……”
Author Response
Many thanks for your enjoyable comments.
Comment 1 to 3: remove ]
Response 1 to 3: We did it
Comment 4: Line 103; Add Tang et al. “Additionally, according to Tang et al. [20]……”
Response 4: we added the name of the author in the text.